

# Estimating flowering transition dates from status-based phenological observations: a test of methods

Shawn D. Taylor

School of Natural Resources and Environment, University of Florida, Gainesville, FL, United States of America

## ABSTRACT

The scale of phenological research has expanded due to the digitization of herbarium specimens and volunteer based contributions. These data are status-based, representing the presence or absence of a specific phenophase. Modelling the progress of plant dormancy to growth and reproduction and back to dormancy requires estimating the transition dates from these status-based observations. There are several methods available for this ranging from statistical moments using the day of year to newly introduced methods using concepts from other fields. Comparing the proficiency of different estimators is difficult since true transition dates are rarely known. Here I use a recently released dataset of in-situ flowering observations of the perennial forb *Echinacea angustifolia*. In this dataset, due to high sampling frequency and unique physiology, the transition dates of onset, peak, and end of flowering are known to within 3 days. I used a Monte Carlo analysis to test eight different estimators across two scales using a range of sample sizes and proportion of flowering presence observations. I evaluated the estimators accuracy in predicting the onset, peak, and end of flowering at the population level, and predicting onset and end of flowering for individual plants. Overall, a method using a Weibull distribution performed the best for population level onset and end estimates, but other estimators may be more appropriate when there is a large amount of absence observations relative to presence observations. For individual estimates a method using the midway point between the first flower presence and most prior flower absence, within 7 days, is the best option as long as the restriction does not limit the final sample size. Otherwise, the Weibull method is adequate for individual estimates as well. These methods allow practitioners to effectively utilize the large amount of status-based phenological observations currently available.

**Submitted** 2 April 2019
**Accepted** 21 August 2019
**Published** 24 September 2019

Corresponding author
Shawn D. Taylor,
shawntaylor@weecology.org

## INTRODUCTION

Plant phenology has a long history in ecological research and is a primary indicator of climate change (*Scheffers et al., 2016*; *Chuine & Régnière, 2017*). Studies commonly document the long-term trends of the first flower or leaf out dates, apply various modelling approaches to infer the drivers of these transitions, or make forecasts using future climate conditions. Phenological models, such as predictive models or those used for long-term trends, use the transition dates as the variable of interest. Common transition dates are

First Observed open flowers or new leaves on a plant, but can also include peak flower, fruit maturation, and leaf senescence. Historic datasets often use repeated observations to identify the true transition date (*Wolkovich et al., 2012*; *Davis et al., 2015*), yet this is susceptible to observer bias (*Miller-Rushing, Inouye & Primack, 2008*). Most modern studies and collection protocols use status-based monitoring, where over time observers record the current state of a single plant (i.e., leaves present or absent) regardless of recent or impending transitions. This includes research using herbarium records, where the presence or absence of flowers and other phenophases is inferred from their presence on a specimen (*Willis et al., 2017*). To make use of status-based data in most phenological models the transition date must first be estimated, and there are several methods available.

Two of the most common estimators are the First Observed and Mean Flowering methods, where either the first observation in a year or the mean dates for all observations within a year is used as an estimate for a phenophase transition and peak dates, respectively (*Miller-Rushing, Inouye & Primack, 2008*; *CaraDonna, Iler & Inouye, 2014*; *Willis et al., 2017*; *Jones & Daehler, 2018*). The First Observed method has been shown to be biased in several instances, while the Mean Flowering date is considered a reliable estimator for the midpoint or peak of a phenophase (*Miller-Rushing, Inouye & Primack, 2008*; *Moussus, Julliard & Jiguet, 2010*; *Bertin, 2015*). Recently more robust methods have been introduced. *Templ, Fleck & Templ (2017)* used survival modelling to estimate the median date of flowering and *Pearse et al. (2017)* used an extinction model to estimate the first flowering date. Using repeated observations of individual plants, as opposed to observations from across a population, site, or region, allows for more reliable estimates. For example if flowers are not present during one visit but present during the next, the transition of flowers opening is constrained to the window between the two visits (*Gerst et al., 2016*). Studies of bird migration phenology face similar challenges and several estimators have been used to model the first arrival dates. Examples include logistic regression (*Mayor et al., 2017*) and General Additive Models (GAMs) (*Moussus et al., 2009*; *Newson et al., 2016*; *Lindén, Meller & Knape, 2017*). To date no comparison has been made of these different transition date estimators for plant phenology.

Furthermore, there are no clear guidelines for using estimators across different spatial scales. Over a latitudinal gradient the transition of a phenophase for a single species can last several weeks to months, and even at the local scale can vary due to many factors (*Diez et al., 2012*; *Zhang et al., 2017*). Studies which estimate transition dates have combined observations from individual plants (*Gerst et al., 2016*; *Taylor et al., 2019*), sites or populations of plants (*Schaber & Badeck, 2002*; *Linkosalo, Lappalainen & Hari, 2008*; *Basler, 2016*), or entire regions (*Calinger, Queenborough & Curtis, 2013*; *Park, 2014*). How different phenological estimators perform across spatial scales is currently unknown.

A comparison of estimators is difficult since, due to infrequent sampling, the true date of transitions is rarely known. Previously *Moussus, Julliard & Jiguet (2010)* used simulated data to test the ability of different estimators to detect shifts in phenological distributions. Here I expand on this prior study by using a dataset of flowering observations from a single population where, due to the unique physiology of the focal species, transition dates can be calculated with high precision, and the efficacy of the different estimators

directly compared. To determine how these estimators perform using different sources of phenological data, such as those from herbarium records or crowd-source applications, I performed this analysis across two different scales (population and individual level transition dates), with varying sample sizes, and with varying proportions of observed flowering presence.

## METHODS

### Phenological data

I used phenological observations of the perennial forb *Echinacea angustifolia* collected in Minnesota, USA in the years 1995–2015 (*Waananen et al., 2018a*; *Waananen et al., 2018b*) to test the accuracy of different transition date estimators. The data consist of the start and end date of flowering (defined as the start and end of pollen production) for 286 individual plants in a 0.5 ha plot from the 11 years of sampling, where the sampling frequency was at least every 3 days during pollen production. The flowering of *E. angustifolia* is such that the true start date of flowering can be inferred very precisely for an individual plant. The flowering heads of *E. angustifolia* consist of 80–250 disk flowers in several rows. The bottom most row flowers first, with each adjacent row flowering every day afterwards. This pattern was used to determine the date of first flowering for an individual to within 2 days for flowering onset and 3 days for flowering end (*Wagenius, 2004*; *Waananen et al., 2018b*). With this information a true start and end date of flowering for the entire population can be approximated.

Different interpretations of phenological metrics can yield different results (*Renzi, Peachey & Gerst, 2019*), thus with the *E. angustifolia* dataset I used strict definitions in calculating the true values used in the analysis. For each year I calculated the following population-level metrics: (1) the start of flowering as defined by the day of year (DOY) of the first observed flower, (2) peak flower defined as the DOY when the most flowers were observed in a given year, (3) the end of flowering as defined by the last DOY a flower was observed. I also calculated two individual level metrics: (1) the start and (2) end DOY of flowering for each individual plant in each year.

To simulate status-based data of a plant population I first determined the flowering status (either present or absent) for every individual plant on every DOY 1–365, then randomly sampled from these dates. Thus an observation could be of flowers present or absent. Flowering absence observations are possible throughout the year as no individual flowers for the full duration of the season. I performed a Monte Carlo analysis, where for every year I repeated this 1,000 times with varying sample sizes (10, 50, and 100 observations) and varying levels of flowering presence being observed (25%, 50%, and 75%). For example, with flowering presence set to 25% using 100 observations only 25 observations were allowed to be of flower presence while the rest were of flower absence, all being randomly chosen from the full calendar year. The variation in sample size and ratio of flowering presence observations simulate patterns seen in non-systematic phenological datasets, such as the those from herbarium records or volunteer contributions. These patterns stem from biases in the time of year of sampling, infrequent or sporadic sampling, or variations in observer effort (*Dickinson, Zuckerberg & Bonter, 2010*; *Willis et al., 2017*; *Daru et al., 2018*).

**Table 1  Estimators used in this analysis.** $p$ indicates the estimator uses only presence observations as opposed to both presence and absence observations.

| | Population | | | Individual | |
|---|---|---|---|---|---|
| | Onset | Peak | End | Onset | End |
| First/Last observed | ✓ $p$ | | ✓ $p$ | ✓ $p$ | ✓ $p$ |
| Midway/Midway 7-Day | | | | ✓ | ✓ |
| Mean Midway/Mean Midway 7-Day | ✓ | | ✓ | | |
| Weibull | ✓ $p$ | | ✓ $p$ | ✓ $p$ | ✓ $p$ |
| Logistic | ✓ | | ✓ | ✓ | ✓ |
| GAM | ✓ | ✓ | ✓ | ✓ | ✓ |
| Survival | | ✓ | | | |
| Mean flowering | | ✓ $p$ | | | |

For individual level flowering estimates I performed the same random sampling routine for every individual in every year using sample sizes of 5, 10, and 20 observations, and flowering presence ratios of 25%, 50%, and 75%. I only used individuals which were in flower for more than 20 days, since below that there would not be enough data in the lowest sample size and flowering presence classes. I repeated this 20 times for each individual in every year in the Monte Carlo analysis.

Each estimator, described below, was fit to each random sample (Table 1). For the population estimates this resulted in 11,000 estimates for each estimator, and sample size/flowering presence combination. For the individual estimates this resulted in 4,840 estimates for each estimator and sample size/flowering presence combination. Estimators were compared using the $R^2$ value between estimated and observed dates of metrics, and by examining the density of errors from all Monte Carlo estimates.

## Estimators

The First Observed method uses the earliest DOY of flowering as the estimate for the start of flowering. Analogous to this is the Last Observed DOY for estimating the end of flowering. These were used in both the population and individual level analysis.

The Midway method uses the midway date between the First Observed flowering date and the most prior observation of flowering absence for an individual plant. This can be improved by applying a restriction whereas only individuals with an observed absence within 7 days of the First Observed presence are used (*Gerst et al., 2016*). Applying this restriction reduces the final sample size available for modelling though. The Midway method was used to estimate onset and end in the individual analysis (Midway and Midway 7-Day), and in the population analysis by using the mean onset date from all individuals (Mean Midway and Mean Midway 7-Day). For all cases I noted the rate at which this could not be calculated due to inadequate sampling (i.e., if no individuals have an absence observation within 7 days prior to the first presence, than no estimate can be made).

The Weibull method fits a Weibull distribution to only the flowering presence observations, thus is advantageous when no absence observations are available. The flexible Weibull distribution can model a variety of shapes, and is commonly used to used to estimate the start or end of a process. The estimated date of first flowering is the sum

of the dates of all flowering weighted by the joint Weibull distribution and is equivalent to estimating an extinction date (*Roberts & Solow, 2003*; *Pearse et al., 2017*). This was used for both population and individual level estimates. Code for this in the R language was obtained from *Pearse et al. (2017)* and is provided in the code repository.

The Logistic method fits a generalized linear model to both presence and absence observations using a binomial distribution, where the DOY was used to explain the presence or absence of flowering (glm(flowering ∼doy, family = binomial)). Prior to fitting all flowering absence observations after the last observed flowering presence were excluded. The expected probability of observing a flower was calculated for all DOYs 1–365, and the estimated onset of flowering was the first DOY in the season in which the expected probability exceeded a given threshold. The inverse of this is used to estimate the end of flowering. All absence observations prior to the First Observed flowering date were excluded, the expected probability was calculated for all DOYs 1–365, and the first DOY where the probability of flowering falls below the threshold was the estimate for the end of flowering. I evaluated a range of probability thresholds (0.05, 0.25, 0.50 ,0.75, and 0.95) and used the one with the highest $R^2$ for each combination of metric, sample size, and flowering presence ratio. This method was used in both the population and individual level analysis.

The GAM method is unique in that it can potentially estimate the full flowering phenology for a season (onset, peak, and end) using smoothing splines. Similar to the Logistic method, a general additive model was fit with a binomial distribution and DOY explaining the presence or absence of flowers, where the DOY was a thin plate regression spline (gam(flowering ∼s(doy, bs = 'tp'), family=binomial)). The expected probability of flowering was calculated for all DOYs 1–365. The estimated onset date was the first DOY in which the probability exceeded a given threshold. The estimated peak flowering date was the DOY with the maximum probability in a given year. The estimated end of flowering was the first DOY, after the peak, in which the probability fell below the threshold. As in the Logistic method I evaluated five probability thresholds and chose the one with the highest $R^2$ for each metric and scenario. Results showing the best probability thresholds for the GAM and Logistic are available in Fig. S4. The GAM method was used for estimating onset and end in both the population and individual level analysis, and for estimating peak flowering in the population analysis.

The Survival method uses a Kaplan–Meier model, which is commonly used to estimate the survival of medical patients. Patient survival (alive or dead) observed in the years following a treatment is used in the model to estimate overall survival probability, with median survival rate, in years, used as a common summary statistic. In a phenology context observations of non-flowering and flowering can be ascribed to alive or dead, respectively, and the DOY, instead of year, of observation used as the time (*Templ, Fleck & Templ, 2017*). The median survival rate can then be interpreted as the median time for flowering. I used the survfit function in the R package survival using right censoring (*Therneau, 2015*). This method was used to estimate peak flowering in the population analysis.

Finally, the Mean Flowering method uses the average DOY of all flowering presence observations from throughout the year. This was used to estimate peak flowering in the population analysis.

All analysis was done using the R programming language (version 3.6.0, *R Core Team, 2017*). Packages used during the analysis included dplyr (version 0.8.1, *Wickham et al., 2017*)), tidyr (version 0.8.3, *Wickham & Henry, 2018*), ggplot2 (version 3.1.1, *Wickham, 2016*), mgcv (*Wood, 2003*, version 1.8.28, *Wood, 2011*), survival (version 2.44.1.1, *Therneau, 2015*), testthat (version 2.1.1, *Wickham, 2011*), ggridges (version 0.5.1, *Wilke, 2018*), and lubridate (version 1.7.4, *Grolemund & Wickham, 2011*). Code to fully reproduce this analysis is available on GitHub (https://github.com/sdtaylor/phenology_estimators) and archived on Zenodo (https://doi.org/10.5281/zenodo.3234913).

## RESULTS

### Population onset estimates

For population level flowering onset the Weibull method produced estimates with the lowest error for most scenarios (Fig. 1). Excluding the scenario where the proportion of flowering presence was 25% and with a sample size of 10, the Weibull method had $R^2$ values from 0.34–0.79 and median error rates of 3–4 days (upper and lower bound errors range from $-20$–$-1$ and 15–8 for the 0.025 and 97.5 quantiles, respectively). With a flowering proportion presence of 25% and sample size of 10 the First Observed method had the highest $R^2$, but still overestimated the true dates by 11 days on average. With higher sample sizes the First Observed method performed comparable to, but always slightly worse than, the Weibull method.

The Logistic and GAM methods had the highest $R^2$, and similar median errors to the Weibull method, when the sample size was high (50–100) and ratio of flowering presence low (25%). In the scenarios where they had the highest $R^2$, the best threshold for estimating onset was 0.25 and 0.50 for the GAM and Logistic methods, respectively (Fig. S4). As the proportion of flowering presence increased, and relative amount of absences decreased, the Logistic and GAM methods tended to perform worse (Fig. 1). This was due to larger time gaps in the data since flowering presence observations occur during a short time window. The gaps resulted in overfit models which increasingly underestimated flowering onset as the proportion of flowering absences decreased (Fig. S5).

The Mean Midway and Mean Midway 7-Day methods were never the best performing methods for estimating population onset. The Mean Midway method did not improve by increasing the sample size or by increasing the proportion of flowering presence observations. Results from the Mean Midway 7-Day method using a sample size of 10 were excluded due to less than 1% of random samples resulting in a usable estimate. This was due to the requirement of each individual plant having at least one presence and one prior absence observation. The usable number of estimates for the remaining scenarios ranged from 2–10% (Fig. S1). With a sample size of 10 the GAM method only produced estimates 27–81% of the time because of too few absence observations, and 100% of the time in all other scenarios.

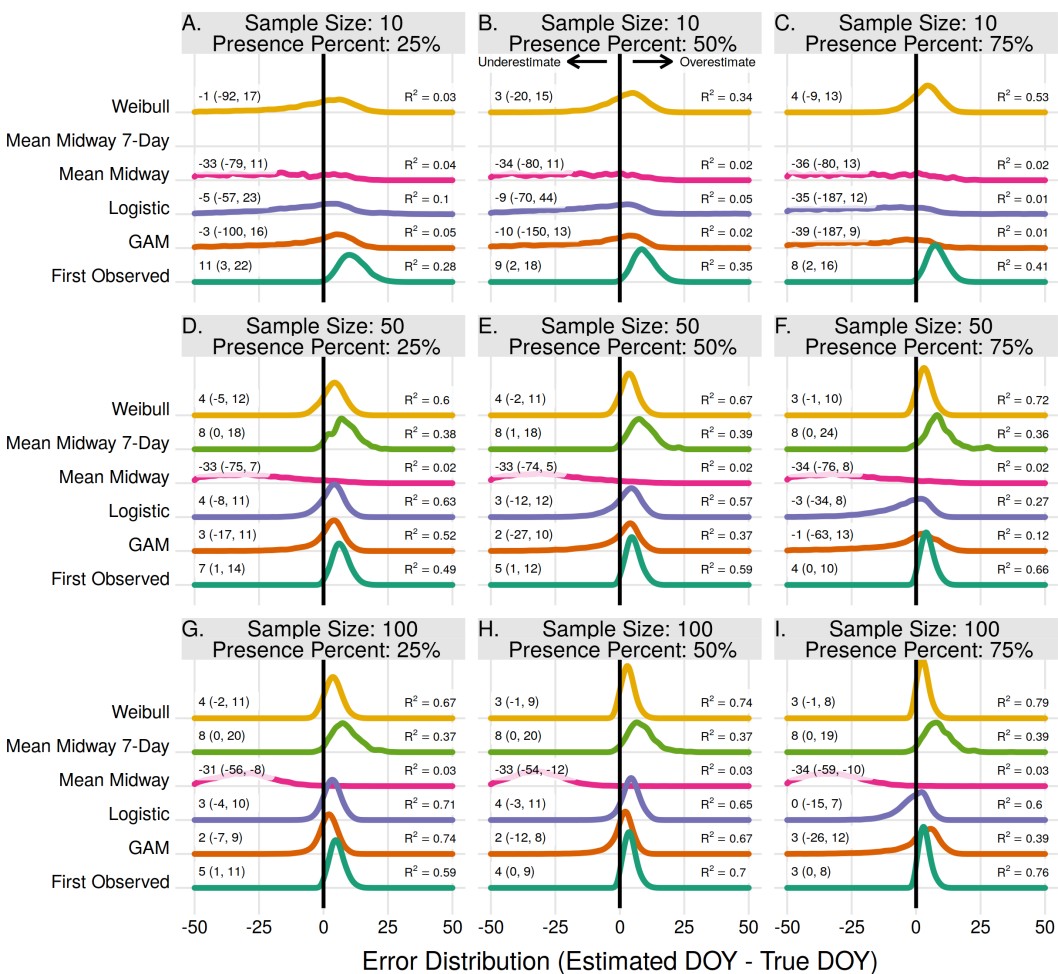

**Figure 1** **The error distribution of all estimators for population onset.** The panels indicate simulated sample sizes of 10 (A–C), 50 (D–F), and 100 (G–I), and presence proportions of 25% (A,D,G), 50% (B,E,H), and 75% (C,F,I). The density curves are each derived from 11,000 randomly drawn observations across eleven years of phenological data. Text values represent the median error and the 95% quantile range in parenthesis.

## Population end estimates

The end of flowering for the entire population was more difficult to estimate than the onset of flowering. The highest $R^2$ for a given scenario in estimating population onset was always higher than the same scenario in population end estimates. For end estimates the Weibull method had the highest $R^2$ in 4 of 9 scenarios, including all three scenarios where the proportion of flowering presence was 75%, as well as when the proportion was 50% with a sample size of 10 (Figs. 2B, 2C, 2F, 2I). With a sample size of 50 and 100 and a presence proportion of 50% and 25% the Logistic and GAM methods had the highest $R^2$ (Figs. 2D, 2E, 2G, 2H). Where it performed the best the Logistic method used a threshold of 0.25 or 0.50 for estimating flowering end, while the GAM method used a threshold of 0.05 (Fig. S4). As in estimating population onset, the Logistic and GAM methods performed worse with increasing flowering presence due to large gaps in the absence data (Fig. S5).
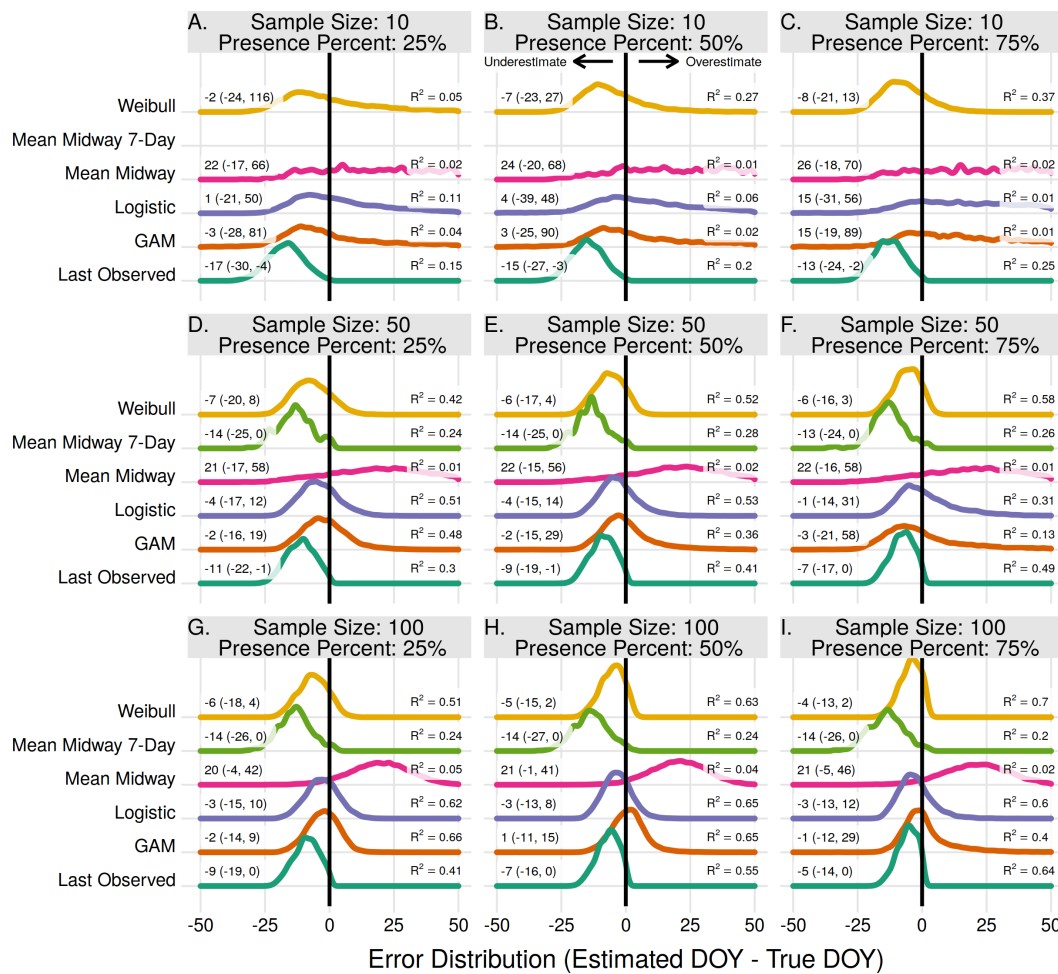

**Figure 2** **The error distribution of all estimators for population end.** The panels indicate simulated sample sizes of 10 (A–C), 50 (D–F), and 100 (G–I), and presence proportions of 25% (A,D,G), 50% (B,E,H), and 75% (C,F,I). The density curves are each derived from 11,000 randomly drawn observations across eleven years of phenological data. Text values represent the median error and the 95% quantile range in parenthesis.

With a sample size of 10 and presence proportion of 25% the Last Observed method had the highest $R^2$, but still underestimated the end date of flowering by 17 days the majority of the time (Fig. 2A). The Midway method, both with and without the 7-day restriction, were never the best performing estimators. Without the 7-day restriction the method consistently overestimated the end date. With the 7-day restriction the method consistently underestimated the end date. Neither Midway method improved with either increasing sample size or increasing proportion of flowering presence. As in the population onset the results from the Mean Midway 7-Day method were excluded due to less than 1% of estimates being usable, and the GAM method had a low proportion (27–81%) of usable estimates with a sample size of 10 (Fig. S1).

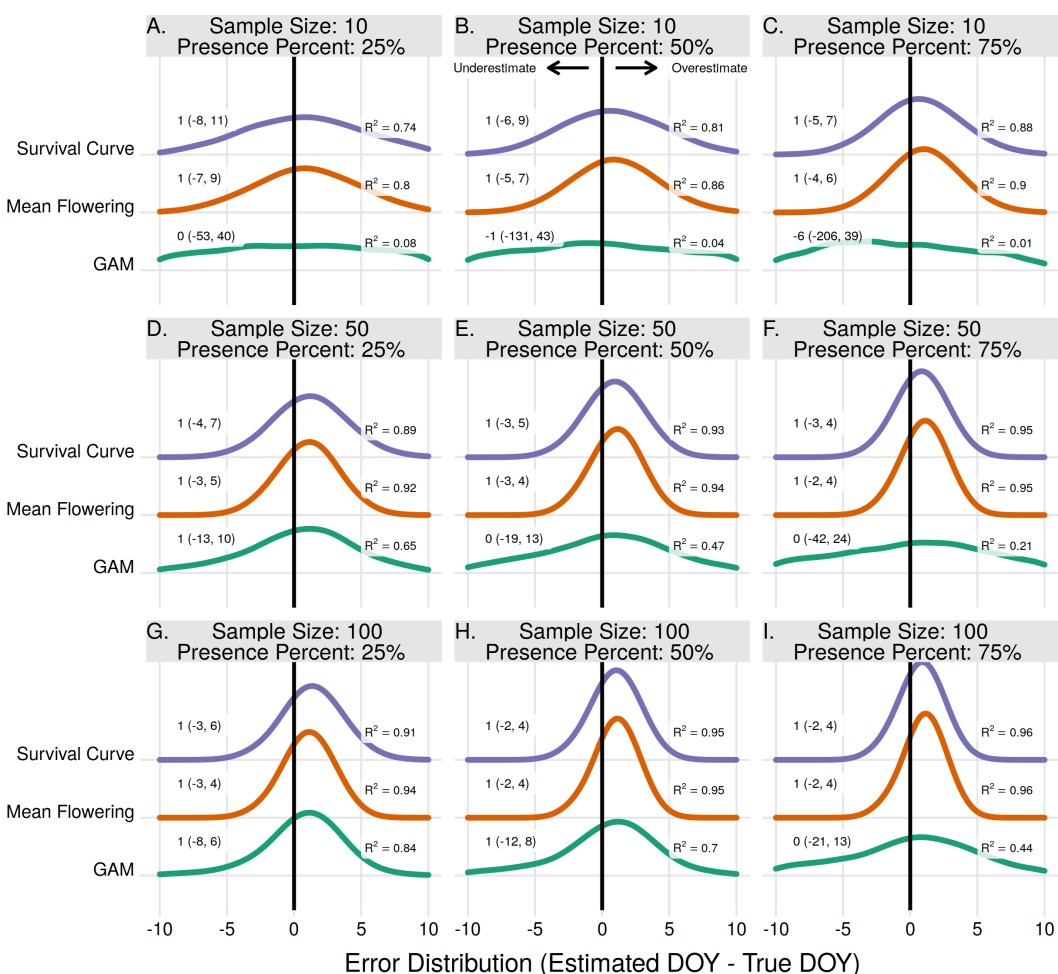

**Figure 3** **The error distribution of all estimators for population peak.** The panels indicate simulated sample sizes of 10 (A–C), 50 (D–F), and 100 (G–I), and presence proportions of 25% (A,D,G), 50% (B,E,H), and 75% (C,F,I). The density curves are each derived from 11,000 randomly drawn observations across eleven years of phenological data. Text values represent the median error and the 95% quantile range in parenthesis.

## Population peak estimates

All three methods to estimate peak flowering had median error rates of 1 day except in one instance, using the GAM method for a sample size of 10 and proportion of flower presence 75% (Fig. 3C). The Mean Flowering method had the highest $R^2$ in all scenarios except three where it had $R^2$ values equal to the Survival Curve method. For the Mean and Survival Curve methods, errors improved with both increasing sample size and increasing proportion of flowering presence. For the GAM method errors improved with increasing sample size, but worsened with increasing proportion of flowering presence.

## Individual onset and end estimates

For individual plant onset estimates the Midway 7-Day method performed the best in 7 of 9 scenarios (Figs. 4A–4E, 4G, 4H). In two scenarios, when the sample size was 15 and 20

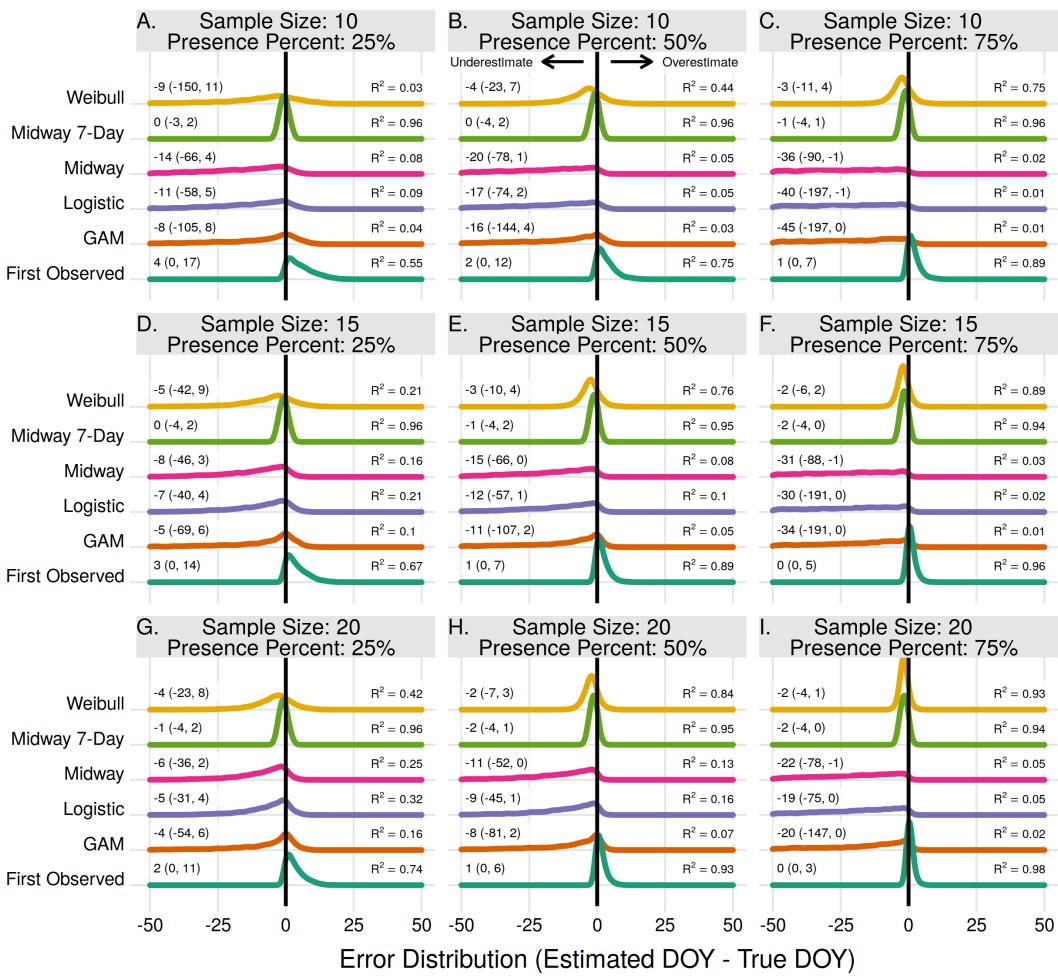

**Figure 4** **The error distribution of all estimators for individual onset.** The panels indicate simulated sample sizes of 10 (A–C), 15 (D–F), and 20 (G–I), and presence proportions of 25% (A,D,G), 50% (B,E,H), and 75% (C,F,I). The density curves are each derived from 4,840 randomly drawn observations across eleven years of phenological data for 286 individual plants. Text values represent the median error and the 95% quantile range in parenthesis.

with a proportion of flowering presence observations of 75%, the First Observed method had slightly higher $R^2$ and lower median error rates than the Midway 7-Day method (Figs. 4F–4I). The Midway 7-Day method was able to produce usable individual estimates only 3–17% of the time due to its restrictive nature, while the Midway method produced usable estimates 70–100% of the time (Fig. S2).

The Weibull, Midway, Logistic, and GAM methods never produced the best estimate for any scenario in estimating individual plant flowering onset. The Weibull method did improve with increasing sample size and increasing proportion of flowering presence. Though, since the Weibull method does not use absence observations, increasing the flowering proportion effectively just increases the sample size. At effective sample sizes of 10 or more the Weibull method produced estimates only slightly worse than the Midway 7-Day and First Observed method. The Midway, Logistic, and GAM methods improved
slightly with increasing sample size but worsened with increasing proportion of flowering presence.

The errors from individual end estimates were nearly identical to individual onset errors, thus the model performance outcomes were the same. Individual end errors are supplied in the supplement (Fig. S3).

## DISCUSSION

### Overall findings

This comparison of phenological estimators using a dataset with known onset, peak, and end of flowering dates confirmed biases in some estimators and shows the strength of newer ones. Overall the Weibull method predominantly outperformed all other methods for estimating the onset and end of flowering populations. The Mean Flowering method produced better, or equal, estimates than other methods for flowering peak. The Midway 7-Day method outperformed other methods in estimating onset and end of individuals flowering, albeit with limitations on the usable sample size. Exceptions to these stem mainly from differences in sample size but also the shape of the flowering distribution.

The Weibull method was the best overall for estimating population onset and end with two exceptions. First, when the total number of flowering presence observations were extremely low (i.e., with a total sample size of 10 and percent presence observations 25%) using just the first or last observed flowering date produced better estimates. Yet with such a low $R^2$ values this method cannot be recommended, and along with other studies I recommend not estimating flowering onset or end with extremely low sample sizes (*Miller-Rushing, Inouye & Primack, 2008*; *Moussus, Julliard & Jiguet, 2010*; *Bertin, 2015*). Second, using a larger sample size (50–100) and a small proportion of flowering presence the Logistic and GAM methods performed slightly better than the Weibull method. This suggests the Logistic and GAM methods effectively utilize flowering absence observations, but require a large amount of them, relative to presence observations, to accurately describe the phenology. Exploring the GAM and Logistic methods further showed that regular sampling, especially during the non-flowering season may also be important. Absence observations are rare in herbarium data due to a bias toward growing season sampling (*Rich & Woodruff, 1992*; *Daru et al., 2018*), but more common in datasets with status-based protocols (*Denny et al., 2014*; *Elmendorf et al., 2016*). Given that flowering absence observations could prove useful when presence observations are low, absence observations should be emphasized in future data collection efforts.

For estimating the peak of flowering populations the Mean Flowering method consistently produced the best estimate, even when the sample size and proportion of flowering presence was low, 10 and 25%, respectively. This method has the advantage over the Survival Curve and GAM method of not requiring flowering absence observations. As noted in other studies the Mean Flowering method is a reliable method for estimating peak flowering (*Miller-Rushing, Inouye & Primack, 2008*; *Moussus, Julliard & Jiguet, 2010*; *Bertin, 2015*).

For estimating the start and end of flowering for individual plants the Midway 7-Day method was the best in most cases. The Weibull method performed similarly when the

absolute number of flowering presence observations was greater than 10, and the First Observed method also performed well with a high amount of presence observations. The First Observed method can be advantageous as it ensures no underestimate of the onset date (or no overestimate of the end date if using Last Observed). In the vast majority of cases (83–97% depending on the scenario, Fig. S2) it was not possible to use the Midway 7-Day method due to lack of individuals with an absence observation within 7 days of the first presence observation. With large enough datasets using this method is still possible even with the restriction (Gerst et al., 2016), and it can also be relaxed with a 15 or 30 day minimum to increase sample size if needed (Taylor et al., 2019). If an insufficient number of individuals results from applying the restriction using the Midway method, then the Weibull or First Observed methods are preferable for estimating onset in an individual given enough flowering presence observations. While the Midway 7-Day method was the best for estimating flowering for individual plants, using the mean of those estimates from a population (Mean Midway 7-Day) did not provide the best population level estimates even with a large sample size. The onset of flowering for individuals is staggered over time and the mean of these start times is not equivalent to the population onset date (Ison & Wagenius, 2014; Keyzer et al., 2017; Renzi, Peachey & Gerst, 2019).

## Prior study comparison

Moussus, Julliard & Jiguet (2010) found GAM's to be among the best estimators for detecting phenological shifts among different seasons, yet here the GAM method performed best only in scenarios with a large proportion of flowering absences. Differences in analysis include Moussus, Julliard & Jiguet (2010) using a Poisson distribution with simulated count data, while here I used a binomial distribution and observed presence/absence data. Moussus, Julliard & Jiguet (2010) also did not evaluate the Weibull estimator, which outperformed the GAM method in many scenerios in the current study. Here the performance of the GAM method was influenced by the proportion of absence observations, where their relative amount affected the best threshold to use as well as the highest accuracy attainable (Fig. S5). Future studies could potentially adjust the GAM model specifications to better accommodate scenarios with a low proportion of absence observations. It is also possible that the output for the GAM model used here, the probability of observing a flower, is not analogous to the total abundance of flowers. General additive models have substantial flexibility (Wood, 2017; Simpson, 2018; Pedersen et al., 2018) and further exploration into their use for plant phenology would be beneficial.

## Drivers of estimator performance

The shape of the flowering distribution affected the proficiency of the estimators. The number of E. angustifolia flowers observed over time resembles a skewed distribution, with a quick onset, peak, and gradual decline in number of flowers. The long tail made end estimates more difficult as the probability of observing a flower close to the true end was low. The best performing estimators also tended to overestimate onset and underestimate end of population flowering, as the majority of randomly sampled observations came from the center of the flowering period. The likeness to a normal

distribution allowed for very accurate estimates of peak flowering using the Mean Flowering method. Flowering distributions for many species are thought to have similar properties (*Forrest & Miller-Rushing, 2010*; *Clark & Thompson, 2011*), but the methods used here may not be appropriate for other phenophases, especially ones which can last significantly longer (i.e., leaves lasting several months on the tree). Flowering is also expected to have non-uniform shifts from changing drivers (*Ogilvie & Forrest, 2017*; *Theobald, Breckheimer & HilleRisLambers, 2017*). Other phenophases which do not have a distinct transition or cannot be easily modelled using presence and absence, such as fruit maturation, may not be well described by the methods used here. In these cases models integrating the continuous cycle of phenology would likely need to be developed, such as using integrated process based models (*Chuine & Régnière, 2017*) or hierarchical bayesian models (*Clark et al., 2014*).

The outcomes for estimating the end of individuals flowering was essentially identical to estimates for the onset. The flowering of an individual *E. angustifolia* plant over time approximates a uniform distribution. Thus, unlike the skewed population flowering over time, estimators for the onset and end of individuals perform equally. This may not be the case when the study species are larger in size and/or contain numerous flowers which can be counted (*Renzi, Peachey & Gerst, 2019*). In these cases the phenology over time may be more similar to a population, with a flowering peak and potentially skewed distribution (while *E. angustifolia*, being in the family Asteraceae, can have one or more flowering heads each with numerous florets, here I treated each individual plant as a single unit).

## Recommendations

Results from this study can be applied to two common sources of large-scale status-based phenological observations, herbarium data and citizen science data. Data from herbarium specimens represent spatially diffuse observations at the population scale or larger, with a bias toward flowering presence (*Willis et al., 2017*; *Daru et al., 2018*). The best onset estimator for these data depends on the type and amount of data available. With a low sample size (less than 10 observations) I recommend not estimating onset as it can lead to high errors (*Miller-Rushing, Inouye & Primack, 2008*; *Moussus, Julliard & Jiguet, 2010*; *Bertin, 2015*). With larger sample sizes the Weibull method will be appropriate in most cases as herbarium data are mostly presence observations, but when there are a large amount of absences the GAM or Logistic methods should be explored. With a very large sample size (greater than 50) the First Observed method can be just as accurate as the Weibull, but note that this accuracy will likely decrease for longer lasting phenophases such as leaves or fruit. For estimating the end of a phenophase the same recommendations apply, with the caveat that the minimum sample size will need to increase if the phenophase distribution has a long tail. As herbarium specimens do not represent repeated observations of the same individual, individual level estimates are not applicable.

Citizen science phenological data can be subset into two types: (1) those from social media applications using geotagged images (i.e., Twitter or iNaturalist, (*Silva, Barbieri & Thomer, 2018*), and (2) those from observing networks and consisting of repeated observations of the same site or individual plant (i.e., the USA National Phenology

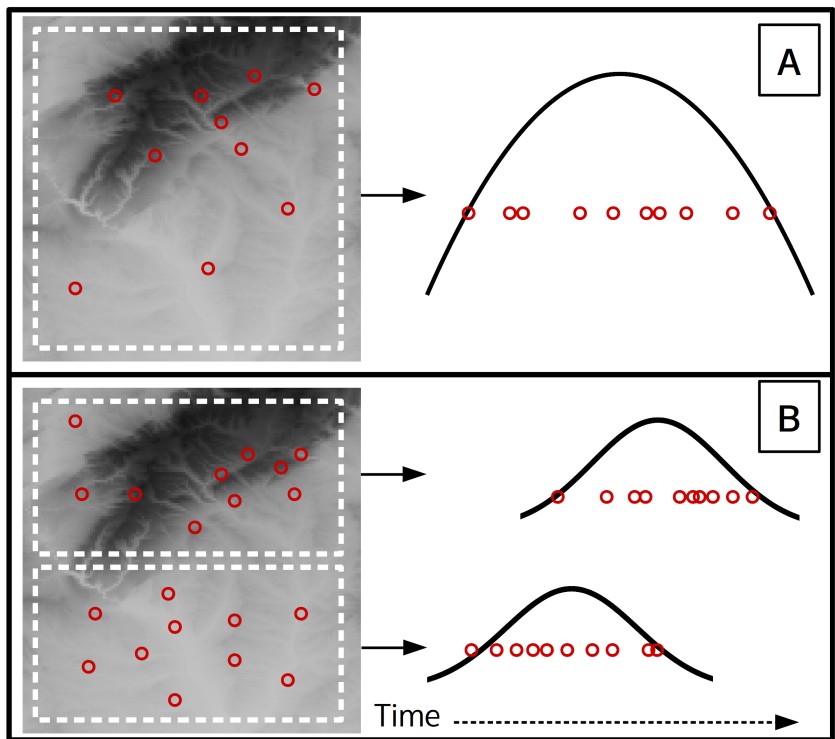

**Figure 5** A theoretical scenario where 10 flowering observations are used to estimate onset across a landscape (A), and a second scenario where onset is estimated at a finer spatial grain on the same landscape (B). Curves indicate the theoretical flowering distribution over time for the respective area.

Network or Pan European Phenological database, *Denny et al., 2014*; *Templ et al., 2018*). For the former the same recommendations as from herbarium specimen data apply. For the latter, if estimates for individual plants are needed then the Midway-7 Day method is most suitable as long as absence observations are available and the final usable amount of data is adequate. Without absences, or to provide more usable data, the First Observed method can be used as long as the sample size is adequate, and the Weibull method should be considered regardless due to its ability to generate confidence intervals (*Pearse et al., 2017*).

This analysis used data from a single site, yet with herbarium or citizen science data observations more commonly represent a large spatial extent. At these larger scales the underlying phenology of a species is affected by an array of biotic and abiotic factors which can cause different flowering times at distant locations (*Diez et al., 2012*; *Keyzer et al., 2017*; *Prevéy et al., 2017*). When combining phenological observations from different locations any transition estimates will be for some subset of the full flowering phenology across the species entire range (i.e., the universal curve, *Keyzer et al. (2017)*). The spatial extent and grain of the analysis will affect the minimum sample size needed and also what the estimates represent due to the modifiable areal unit problem (*Jelinski & Wu, 1996*). For example consider a case where 10 observations of flowering from a single year are used to estimate onset (Fig. 5A), which represents flowering onset for the entire landscape. If the same landscape is subset to a finer spatial grain (Fig. 5B), then each of the two

smaller spatial units could have an independent onset estimate, but would each require an adequate sample size. Also note that the onset estimate for the larger grain (Fig. 5A) will likely approximate the onset estimate of the earlier of the two smaller grain estimates, while the larger grain end estimate will approximate the later of the smaller grain estimates. Previous studies used political boundaries as the spatial unit (*Park, 2014*; *Pearse et al., 2017*), though the optimal spatial grain and observation density needed likely depends on the species being analysed and the large-scale gradients over which it occurs. Future studies should examine these relationships between spatial scales and phenology more closely.

## CONCLUSION

In summary I have used a precise flowering phenological dataset to confirm that naively using the first flowering observation is biased, and estimates using the Mean Flowering reliable for estimating flowering peak. I have also shown how the recently introduced Weibull method can produce reliable estimates given an adequate sample size. The Logistic and GAM methods can be useful with large datasets having low amounts of flowering presence, and future collection efforts should emphasize absence observations for this reason. Additionally, estimating transition dates of individual plants is best done with the Midway method using a 7 day restriction, and the Weibull method if the restriction results in a low number of final samples. These estimators are needed for translating status-based phenological data into distinct transition dates used to track changing seasonal patterns.

## ACKNOWLEDGEMENTS

I thank Amy Waananen and the team of the Echinacea Project (http://echinaceaproject.org/) for providing the phenology data set. I thank Amy Waananen and Janet Prevéy for providing feedback on early versions of the manuscript.

### Funding
This research was supported by the Gordon and Betty Moore Foundation's Data-Driven Discovery Initiative through Grant GBMF4563 to Ethan P. White. The funders had no role in study design, data collection and analysis, decision to publish, or preparation of the manuscript.

### Grant Disclosures
The following grant information was disclosed by the author:
Gordon and Betty Moore Foundation's Data-Driven Discovery Initiative: GBMF4563.

### Competing Interests
The author declares there are no competing interests.

## Author Contributions

- Shawn D. Taylor conceived and designed the experiments, performed the experiments, analyzed the data, contributed reagents/materials/analysis tools, prepared figures and/or tables, authored or reviewed drafts of the paper, approved the final draft.

## Data Availability

The code to fully reproduce this analysis is available on Github (https://github.com/sdtaylor/phenology_estimators), and archived, along with the data, on Zenodo:

Taylor, Shawn. (2019, March 29). Code and data used in the study: Estimating transition dates from status-based phenological observations: a test of methods. Zenodo. http://doi.org/10.5281/zenodo.3234913.

## Supplemental Information

Supplemental information for this article can be found online at http://dx.doi.org/10.7717/peerj.7720#supplemental-information.

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
