# Peer review of "Estimating flowering transition dates from status-based phenological observations: a test of methods"

_PeerJ, doi:10.7717/peerj.7720_

## Round 0.1 · original submission · Minor Revisions

This is a very useful paper that spells out many different approaches to estimating phenology and also provides a very useful comparison of these methods.

The reviewers liked the paper and thought it would be very useful, but noted the limited scope. I think that you could be more up front about the limits of this research. You say in the discussion that the methods and results really apply to a fairly narrow circumstance, a single site that is being monitored regularly, whereas that is rarely the structure of incoming data for phenology studies.

Please pay special attention to reviewer #3 who makes several suggestions about increasing the usefulness of the provided code. Both reviewer 1 and 3 offered several suggestions for tightening the narrative.

Note that there was some confusion about how you were counting plants vs. individuals. I think more information about the patch that was being monitored would be helpful. How many individuals (however you count those) were monitored each year. Presumably all individuals within a defined patch were monitored. Also, since you were considering the phenology of all the disc flowers within a composite species, it wasn't clear to me if that would impact the generalizability of the results (does it influence the pattern/length of flowering and thus the detectability of flowering in this species compared to others? Please address that issue.

Finally, I noted that some of your simplest metrics were very high performing compared to the Weibull. Is it worth mentioning that, at least within this set of circumstances, simple metrics may suffice to capture a fairly accurate date of flowering?

Overall - a very useful study.

Reviewer 1 ·

Basic reporting

General Comments: This is a well thought out, useful paper that that is generally well-written, and will be of interest and practical utility to phenological researchers in the future.

Experimental design

This experiment seems well designed, although there are a few places where aspects of it could be better communicated (see comments to the author)

Validity of the findings

These findings seem legitimate.

Additional comments

Lines 15-17 since you bring up herbarium data earlier, it might be helpful to mention that this is an in-situ dataset for purposes of clarity.

Lines 19-21 It seems a little odd to just say you tested estimates, when the paper is testing the accuracy of estimates produced through multiple different methodologies. This absence may be a reflection of the word limit associated with the abstract, but some mention that you are testing estimates constructed using multiple methodologies would be appropriate here.

Line 35 would ‘phenological’ be more appropriate than ‘phenology’ here, since it’s being used as an adjective?

Line 66 This is a minor point, but if you’re referring exclusively to spatial scales, it might be best to say that explicitly – phenological studies can also exist across scales not only spatially, but also temporally and at individual – population levels.

Line 84: what did the phenological observations consist of? Clearly they record more than simple flowering/nonflowering, but a little more explicit explanation would be helpful. (i.e., are the number of disk flowers within a head recorded, or the proportion? How many flowering heads were recorded per individual, and what was the protocol for selecting them, or handling cases where the proportion of open disk flowers differed among heads?) Also, I think ‘phenological’ might be more appropriate than ‘phenology’ since it’s a descriptor.

Line 91-92 – to what accuracy can the date of first-flower be back-estimated? How was this tested? If the accuracy wasn’t explicitly tested I don’t think that’s an insurmountable problem, but if you’re claiming total precision (as seems to be the case here) then some explanation of how you can be confident that’s the case seems appropriate.

. Lines 101-103 - This statement is a little ambiguous to me. Did you randomly select a subset of the actual observations of each individual (which are not made every day, and are unlikely to be made during parts of the year when the plant is unlikely to be in flower), or did you, knowing the actual dates of bloom onset and termination, randomly select a number of dates throughout the year (on which an actual observation may or may not have occurred) and then populate them with what you can infer the phenological status was using the actual observations? Based on the following text I believe it’s the latter. If so, it needs to be more explicit, as my initial reading from this statement was that it was the former.
. Lines 106 “observations”
. Line 142 some brief description of the Weibull distribution might be helpful here, as the reader may not be familiar with it. For example, this particular distribution has a number of advantages over several other probablility distributions in that it can be quite flexible in its shape, or handle skewed distributions.
. Lines 146 and 147 – it’s probably appropriate to note what language system the code is for in the main text (i.e., python, R, etc). Also, it might be best to include the web address or DOI for the code repository here.
Lines 150, 163 - it seems odd to have these orphaned lines of R code inserted into the main text. I would either simply state that you used this method in R using XYZ package, or include the full code in the supplement and then refer to that in the main text.
Lines 178-182 – For form’s sake, it’s probably best to include the version of R (and of the relevant packages) that were used in this study.
. Line 290 – It may also be worth noting that many herbarium specimens do not explicitly document phenological status, even within herbaria that typically do record phenology – thus, while specimens recorded as in flower can be assumed to be in flower, specimens not recorded as being in flower (which are often the majority of specimens) may potentially be either in flower or not, and cannot therefore be reliably used as observations of flowering absence unless accompanied by imagery or a manual assessment of flowering status
. Line 322 replace midway with midiway
. Discussion – I think some discussion of the applicability of each method to different kinds of existing phenology datasets would improve the utility of this manuscript. For example, while the midway-7 day method may perform best in some cases, it may not be usable with datasets such as herbarium specimens, in which absence observations are not present or are unreliable. Conversely, this would further underscore the utility of the Weibull method, as the fact that it does not utilize observations of absence makes it a better fit for studies using those types of phenological data that do not provide them. Emphasizing that it not only performs best in many cases, but also is a better fit to the nature of many phenological datasets, would help communicate the importance of this finding.

·

Basic reporting

I suggest editing text in the last paragraph of the introduction where you introduce what you did and why and set up the methods - I think you need a sentence linking the sentences on line 76 and 77, and a statement of the overall objective of the study. I also suggest adding subheadings in the discussion section for readability and flow.

Experimental design

No comment.

Validity of the findings

No comment.

Additional comments

Overall, this is a great contribution to the phenological literature, as there is an abundance of methodologies used to generate the unit of measurement across many disparate datasets and this provides some “best practices”. The synthesis presented in a very useful and straightforward manner, and the methodology is innovative and robust. Would be interesting to follow up with a comparison of peak flowering at the individual level. I’d also suggest including context in the discussion of when someone might choose to use individual estimators in comparison to using peak estimators – for decision making in data analyses and applications.

A few small thoughts: I suggest not using the term “Julian day” for the day of year, as Julian day technically refers to the number of days since Jan 1, 4713 BC. [Or clarify that you are using a specific definition of this term that restricts DOY values between 1-365]. I would also clarify that the “mean occurrence day” is not the mean of all the first observed DOYs but rather the mean of all the positive records for a year. I’d suggest citing Renzi et al. 2019 American Journal of Botany for a similar (but less detailed) look at population vs individual level metrics. I’d suggest using the term “metrics” rather than “estimators”; as it was somewhat confusing to go back and forth between the term estimates (of onset and end dates) and estimators.

Reviewer 3 ·

Basic reporting

The reporting in the article is unambigous and professional English throughout. However, I do suggest to make sure that all figures have a full caption without cross references.

Experimental design

The research is well defined, and full access to the code and data is provided. The code runs. I did not check for bugs or other calculation errors as I consider code review beyond the scope of a manuscript review. I assume good faith and relative stability of the code provided.

Validity of the findings

The study is a meaningful addition to previous work, in particular Moussus et al. 2010 / Pearse et al. 2017, while adding transparency in code and data used. I would include some stronger (broader) statements providing the context provided by these previous studies (see general comments).

Additional comments

The manuscript titled "Estimating transition dates from status-based phenology observations: a test of methods" compares various approaches to characterize and estimate true transition dates from status based observations.

Status based observations dominate various phenology records not in the least those derived from herbarium specimen, but also and certainly not less important in a contemporary setting, crowd-sourced data. All these data sources are important in understanding past and current phenology responses (and their interactions). A better understanding of the influence of how data is processed and its influence on error distributions is a valuable addition to ongoing work in the field of (vegetation) phenology.

I think this study certainly merits publication. However, I feel that the scope has been left too narrow in either defining the question and the final conclusion. This is more a matter of how the text is structured rather than any flaws in referenced literature. I would encourage the author to look for ways to improve the narrative, and in generally make the manuscript more concise.

Reading through the discussion I feel that there was some repetition involved and this section might benefit from some tightening up and more concise writing. While references to common data sources such as the PEP725 dataset, USA-NPN data or other citizen science based data is sparse. Given the scope of the analysis I think some more context in this respect would benefit readers to have an overview of some of these (and other data sources), if not in a small overview table.

I would rephrase the language in the final concluding paragraph, and as a lead up in your introduction, to be a bit stronger (especially when combining the current results with those of other studies, i.e. Moussus et al.). I think it is fair to state that in large but sparse datasets (low presence large absence data) either Weibull or GAM methods merit consideration. Their should be caution when dealing with smaller datasets (which might be common if not the norm in certain contexts, it might be beneficial to highlight those). Absence values are of importance, and hence should be gathered where and if possible. This is mentioned in passing in the text, but I would argue that repeating this message again in a conclusion as it would benefit the design of future studies. The presence of the analysis code also ensures that users can evaluate their methodology up front before preceding with additional analysis which should limit future ad-hoc decisions. I think this is one of the main advantages of open science, and therefore should not be undervalued! However, some work might be needed to make the coding framework a bit more transparent (see below technical comments).

I recognize that the limitation of using one species, which can't be used to generalize, the availability of code and data ensures scalability of this analysis to other data sources. In this context I'm missing an explicit reference to a code and data repository in the methods or a data availability statement. This would help people find code and data quicker, and repeat the analysis on their own data if so desired (again more on this in the technical comments).

Given the above comments, and some minor details below I think this should make a nice reproducible paper. I can recommend a minor revision and wish the author all the best with implementing these changes.

Technical comments:

I've ran the code, although it did not complete (longer computational time, and my impatience). It might be useful to include a README with details on:

1. the system requirements (R packages mainly)
2. how to use the code (what to run first)
3. what to expect in terms of processing time.

My suggestion would also be to create a formal R package as there is no file outlining package requirements. A formal R package DESCRIPTION file (analogues to a python requirements.txt file) would limit issues regarding missing packages before running the code (and frustration that might come with it for novel users). Alternatively, I encourage the author to include an R script which checks if all requirements are met.

As mentioned above, the strength of an open publication in all its aspects (manuscript + code + data) lies in the fact that the analysis ideally can be run on different datasets with ease in order to limit future ad-hoc decisions. I think this idea should be kept in mind when providing code.

Minor comments:

L16 Be explicit about the species / dataset name, this seems to be ommitted but rather key information even for an abstract.

L21 Mentions 'recently' twice in short succession, might want to look for an alternative.

L22 Why is the Weibull curve method new and what are the advantages? Add a line with this info to the abstract as to get people up to speed. Repeat this line in the method section as well.

L232 Is the described pattern a form of overfitting?

L250 I suggest to combine the individual estimates (onset / end) into one section.

L281 Absence data might be as valuable as presence data, how is this currently implemented in observations schemes in particular citizen science components (USA-NPN etc.) (resolved in L293 - but might need consolidation)

L273 How does this observation reflect on the use of these statistics on herbarium data? I suspect they all might have low(er) observation numbers?

---

## Round 0.2 · Minor Revisions

Please attend to the following minor comments. In general, both myself and one other re-reviewer were pleased with how you addressed the concerns of the previous reviews.

See reviewer 1's minor comments.

I add one more request to have an even stronger clarification of the scope of these results. Your study only applies to flowering phenology, so I request that you include that in your title, e.g., "Estimating flowering transition dates from status-based phenological observations: a test of methods".

A couple of other suggestions:

When comparing your results to Mousus - can you say whether or not they used the weibull method (I'm assuming they didn't) - but it is important to put those results in that context.

I found the survival analysis method to be a bit confusing. Can you clarify that a bit more. Why does it only estimate peak flowering time?

Absent further work where you can examine how variability in flowering date across a population would impact the required number of observations from "presence" of flowering records, can you give any guidance of how big a spatial scope should be used to get what appears to be a recommended number of 10 flowering events (that is presumably all within a single year?). This seems to be a key outcome of your paper - that one needs about 10 observations to do an analysis using herbarium or iNat-like data (and I'm sure you'd make any recommendations with care, but people do tend to latch onto these thresholds).

Can you think of a back-of-the envelope way to give any kind of spatial recommendation for how big an area is appropriate to try to reach that 10 observation threshold? Can you do an analysis on NPN phenology maps to see how large a grid cell can be before there is too much variablility to reliably estimate greenup (or whatever) dates within a single pixel?

·

Basic reporting

No comment.

Experimental design

No comment.

Validity of the findings

No comment.

Additional comments

The author did a very nice job responding to the reviewers' comments and improving the manuscript substantially. This paper will be a very useful contribution to the field of phenology and help guide researchers in assessing the best approach for selecting methods to calculate transition dates given their dataset limitations and structure.
I only have a few *very* minor edits/suggestions below:
Line 40: change "leafs" to "leaves"
Line 46: clarify you mean repeated observations on the same plant(s).
Line 51; sometimes "First Observed" method is capitalized and other times not, check throughout the manuscript for consistency; also clarify here if you mean these estimators to be for individual or population, or both.
Line 54: I think this statement requires additional citations for context.
Line 81: include that this is for one site/population.
Line 85: could better explain how "population" is defined in the context of these results, some might use these methods to estimate dates across multiple sites/regions. For example, in studies that rely on herbarium data, how are populations defined, and are there often many samples from one "population"?
Line 139: should it be "Last Observed"?
Line 145: by how much was final sample sizes reduced in this study when this restriction was applied?
Line 166: any guidance regarding what a reasonable threshold would be?
Line 209: what is the range of error rates?
I really appreciated the recommendations section of the discussion- very comprehensive and straightforward.

---

## Round 0.3 · accepted · Accept

Thanks for your careful attention to the comments - and I look forward to seeing this paper published. Great job!